
# Community Based Early Warning Systems for flood risk mitigation in Nepal

Paul J. Smith[1], Sarah Brown[2], and Sumit Dugar[3]

[1]Lancaster Environment Centre, Lancaster University, UK
[2]Practical Action Consulting, Rugby, UK
[3]Practical Action Consulting, Kathmandu, Nepal

*Correspondence to:* Paul Smith (paul@waternumbers.co.uk)

**Abstract.** This paper focuses on the use of Community Based Early Warning Systems for flood risk mitigation in Nepal. The first part of the work outlines the evolution and current status of these community based systems. A significant ongoing challenge faced by Community Based Early Warning Systems in Nepal is the short lead times available for early warning. The second part of the paper therefore focuses on the development of a robust operational flood forecasting methodology

for use by the Department for Hydrology and Meteorology (DHM), Government of Nepal to compliment the community based systems. The resulting methodology uses data based physically interpretable time series models and data assimilation to generate probabilistic forecasts. The paper concludes with an example application to a flood prone catchment (Karnali Basin) in western Nepal.

## 1   Introduction

Nepal is considered one of the most disaster-prone countries in the world (NRRC, 2011). Alongside other natural hazards such as earthquakes and landslides flooding posses a ongoing, annual, risk to large sections of the population. Between 1971 and 2011, 3,520 flood events were recorded, causing 3,329 deaths and affecting 3.9 million people (DesInventar (n.d.)) More recent events such as the 2013 Mahakali catastrophe (Regmee et al., 2014) and flooding the Karnali basin (Fig. 1, Sect. 5 and Zurich, 2015) continue to impact on the livelihoods of the populous; though in part due to the methods outlined in this work the loss

of life has been substantially decreased.

Detailed analysis of two of these events; the 1993 Central Nepal floods (NCVST, 2009) and the Koshi embankment breach in 2008 (Dixit, 2009) highlights the extent to which flood risk results not just from the magnitude of hazard but as importantly from the vulnerability of the population. Vulnerability stems from multiple sources (Khanal et al., 2015) but is often significantly increased by a low level of preparedness on both the individual and institutional level resulting in mitigating actions being

taken too late if at all. Section 2 outlines in more detail the nature of the flood risk in Nepal.

Improving the flood resilience of both the people and infrastructure in Nepal to floods is a priority for the national government (NRRC, 2011; MoHA, 2015) and the international community. This has led to various initiatives such the World Bank funded 'Building Resilience to Climate Related Hazards' program (http://brch.dhm.gov.np/) as well as involvement in region institutions such as the 'Regional Integrated Multi-Hazard Early Warning System for Africa and Asia' (RIMES,



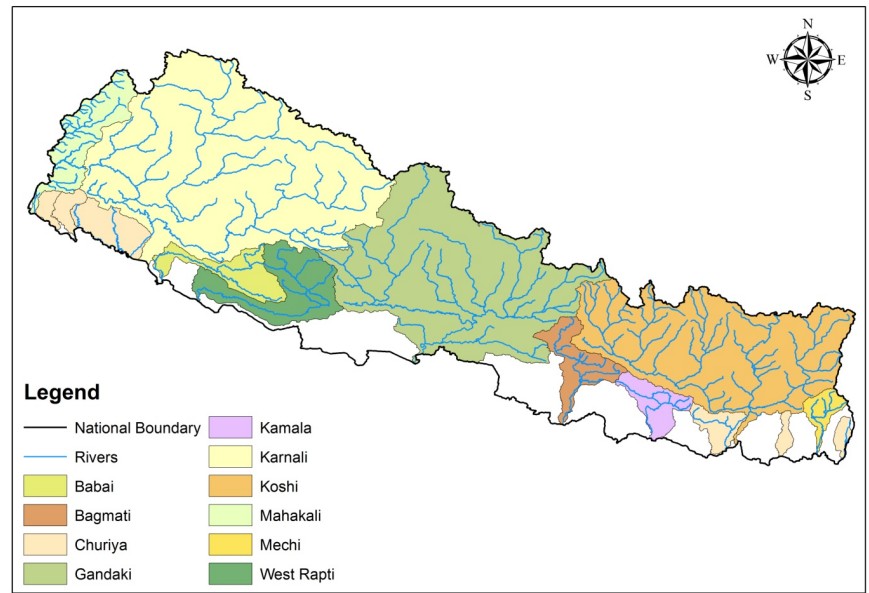

**Figure 1.** Map of Nepal showing the major river basins

http://www.rimes.int). Both these initiatives collaborate with the Government of Nepal; particularly the Department of Hydrology and Meteorology; to enhance the institutional capacity to anticipate future floods through activities such as expanding the hydrological and meteorological station network or development of prototype forecasting systems.

An approach to improving resilience, the development of which has been supported by International Non-Governmental
Organisations, is the use of Community Based Early Warning Systems. In 2002, Practical Action piloted a Community Based flood EWS in East Rapti/Narayani river basin in Nepal. Community Based Flood Early Warning Systems are in place in a wide range of countries, for example in Malawi with support from Christian Aid (Brown, 2014) and in Indonesia and Cambodia with support from national societies of the Red Cross Red Crescent (IFRC, 2010). In Kandal province, Cambodia, local authority and Red Cross volunteers regularly monitor and post updated information on water level on a display board during the flood
season (IFRC, 2010) whilst CBEWS in Indonesia has been trialed in Letye in the Binajaan river basin (Practical Action and Mercy Corps, 2012). Practical Action is also working with local NGOs in both Nepal and India on cross border EWS across the Ghagra river basin (Karnali in Nepal) in India in Uttar Pradesh (Shukla and Mall, 2016; Practical Action and Mercy Corps, 2012).

Section 3 outlines the evolution, current status and ongoing limitations of Community Based Early Warning Systems (CBE-
WSs) in Nepal. While successful at saving lives the time given for action, typically two-six hours, means that CBEWSs are not adequate for saving assets or livelihoods (Zurich, 2015). Increasing the time given for action while maintaining the accuracy and precision of the warning information would allow for a more pro-active response to flood risk. To this end Section 4 fo-





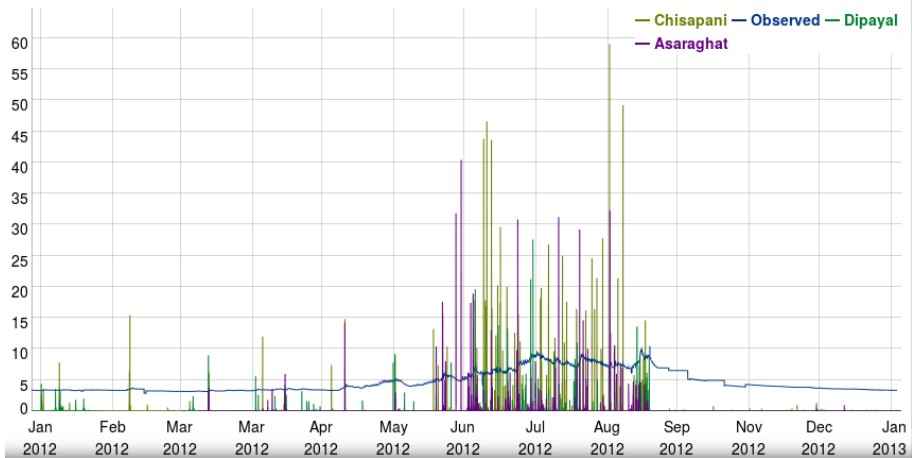

**Figure 2.** An example year of data for the Karnali basin showing the observed water level at Chisapani (observed) and rainfall recorded at three gauges (Chisapani, Dipayal and Asaraghat). Note the spatial variability in the rainfall and, even in this comparatively dry monsoon the river range of approximately 7m

cuses on the development of a robust operational flood forecasting system to enhance early warning and disaster preparedness in Nepal. Section 5 outlines a pilot application to the flood prone Karnali basin in western Nepal.

## 2 Flood Risk in Nepal

Nepal is bordered by the Himalayas in the north and Indian plains to the south (Shrestha et al., 2008). The country has a diverse
topography which can be classified into five physiographic zones extending from the east to the west of the country (NCVST, 2009). From south to North these are the Terai plains, Churia hills (Siwaliks), Middle Hills, High Mountains and the Himalyas (Zurich, 2015).

Figure 1 shows the main river in Nepal. There are both major snow-fed rivers in the Kohsi, Gandaki, Karnali, Mahakali and mid size rain-fed rivers such as Mechi, Kankai, Bagmati, East Rapti,West Rapti and Babai. Intermittent rivers originating from
the Siwaliks are subject to frequent flash floods and carry high sediment load (Sharma, 1997), despite having no significant flow in non-monsoon seasons.

The climate regime of Nepal is mostly affected by the monsoon (June to September) and westerly circulation systems, with the former being dominant (Gautam and Phaiju, 2013). Analyzing the historical records made available by the Department of Hydrology and Meteorology (DHM) indicates that the 70-80% of total river flow occurs during the monsoon (Gautam and
Phaiju, 2013) whilst snow melt contributes to pre-monsoon and post-monsoon stream flow. Similarly, the South Asian monsoon on average also accounts for 80% of annual rainfall (Shrestha et al., 2014). The concentrated period during which river flows occur result in marked changes between in flow and water level in summer and winter. For example the changes in water level shown in Fig. 2 represent a change in discharge from around 500 m³/s to 10000 m³/s.



Flooding is generally ubiquitous during the monsoon in Nepal (June to September) and mostly impacts the vulnerable communities residing in the Terai floodplains (NCVST, 2009), affecting their livelihoods, particularly subsistence agriculture in the floodplains (Zurich, 2015). As well as damage to lives and property, floods damage critical roads, communication infrastructure and power supply, significantly impacting development (Gautam and Dulal, 2013). Floods kill livestock of critical importance to poor communities, with 14,571 cows, pigs, chickens and goats killed in the 2008 Koshi flood (Baral, 2009). Floods deposit sand on farmland, negatively affecting agricultural livelihoods and food production (Gautam and Dulal, 2013).

Despite the impact of flooding the highly variable magnitude, duration and intensity of precipitation at the macro and micro scales across the country during the monsoon (NCVST, 2009) coupled with limited understanding of the response of river flows to the high intensity and short duration precipitation events which trigger floods downstream (Shrestha et al., 2008) makes forecasting challenging. The production of forecasts is further complicated by the dynamic geomorphology coupled with topographic and geological constraints which make the collection of reliable data difficult (Nepal et al., 2014).

## 3 Community Based Early Warning Systems

Early Warning Systems (EWS) are defined as "the set of capacities needed to generate and disseminate timely and meaningful warning information, to enable those threatened by a hazard to prepare and act appropriately and in sufficient time to reduce the possibility of harm or loss" (UNISDR, 2006). Early Warning is a key component of Disaster Risk Reduction. Enhancement of risk monitoring and early warning is the second priority of the Hyogo Framework for Action 2005-2015 (UN, 2005), and a key component of the current Sendai Framework for Action 2015-2030 (UN, 2015).

The Hyogo Framework prioritised the development of People Centred Early Warning Systems (EWSs) encompassing four critical components: risk knowledge; monitoring and warning; dissemination and communication, and response capability (UNISDR, 2006; UN, 2005; IFRC, 2010). Effective EWSs provide warnings that are accurate, timely and understandable, enabling at-risk groups to take appropriate responses (Shrestha et al., 2014). Weakness in any of the four key areas can result in EWS failure (UNISDR, 2006; Kundzewicz, 2013).

People Centred, or, Community Based Early Warning Systems (CBEWSs), such as those outlined in this paper, help communities use local resources and capacities to effectively prepare for, and respond to, hazard events This enables communities to reduce their vulnerability to floods and other hazards (Mercy Corps and Practical Action, 2010).

### 3.1 Community Based Early Warning Systems (CBEWSs) in Nepal

The first Nepalese CBEWS was piloted by the NGO Practical Action in 2002 for the East Rapti River in Central Nepal (Practical Action, 2008). This initial pilot has been enhanced and expanded, and CBEWSs are now operational in eight river basins across Nepal (Karnali, West Rapti, Babai, East Rapti, Narayani, Bagmati, Kankai and Koshi basins) (Gautam and Phaiju, 2013). These CBEWSs consider all four key EWS components listed in the Hyogo framework as will be outlined below.



### 3.1.1 Risk knowledge

Risk knowledge involves assessing and mapping key hazards, vulnerabilities and exposure (UNISDR, 2005). At community level, this information often comes from Community Risk Assessments (IFRC, 2012). In CBEWS an emphasis is placed on community actors having sufficient awareness, information and understanding of risk (vulnerability and hazards) (IFRC, 2010; Mercy Corps and Practical Action, 2010). Early CBEWSs in Nepal worked directly with communities to map historic flood events and to determine the relationship between observed river height at an upstream location and expected inundation downstream. Upstream 'warning', and 'danger', water levels were established for pilot river basins, based on past floods (Gautam and Phaiju, 2013).

### 3.1.2 Monitoring and warning

This component includes hazard monitoring, defining parameters and indicators on which to base early warning, and ensuring accurate and timely forecasts and alerts (UNISDR, 2005). Flood Early Warning Systems commonly monitor rainfall and river height (Shrestha et al., 2014). Early CBEWSs in Nepal trained community gauge readers to monitor river levels and disseminate warnings to downstream communities when river levels rose above defined thresholds (Mercy Corps and Practical Action, 2010). Community level structures such as Community Disaster Management Committees, monitor and record information on flood levels, duration, and impact (Gautam and Phaiju, 2013). In CBEWSs communities are the active owners and drivers of their EWS, engaged in monitoring and, or, analysis, rather than passive receivers of early warning (IFRC, 2012).

Over the past decade, and in close partnership with the Nepal Department of Hydrology and Meteorology (DHM), these community systems of manual river-gauge monitoring have evolved to become integrated with national systems (Practical Action and Mercy Corps, 2012). DHM now has 286 meteorological stations nationwide, and 170 hydrological stations (Shrestha et al., 2014). Most of these stations are manually operated, though some have been upgraded to automatic stations, with continuous monitoring of water level and rainfall. This monitoring network links to a DHM managed web-based telemetry Flood Early Warning System, with real time data publically accessible through the DHM website (Shrestha et al., 2014). Data is transmitted to the DHM server every 5 minutes, with flood warning bulletins available on www.hydrology.gov.np throughout the monsoon period (Gautam and Phaiju, 2013).

### 3.1.3 Dissemination and Communication

Dissemination and communication focuses on whether warnings reach all at-risk groups, whether such warnings are understood, and if they are acted upon (UNISDR, 2005). Effective early warning messages convey timing (when the hazard is due to strike); location; scale; impact (what will be the effect on at risk groups); probability and response (what should at-risk populations do to protect themselves) (IFRC, 2012). Systems are put in place and tested, to ensure early warning is disseminated widely, in a timely and efficient manner (WMO, 2015). "An actionable early warning provides a timely message that reaches, is understood and is acted upon by the population at-risk" (IFRC, 2012).



In Nepal, gauge readers carefully monitor when water levels approach warning and danger levels, ready to disseminate warnings (Mercy Corps and Practical Action, 2010). District-level government maintain electronic flood monitoring display boards, with sirens that sound automatically when water reaches warning levels. The web-based telemetry system also triggers a SMS, text message, warning that is sent to Chief District Officers (Gautam and Phaiju, 2013). Building such redundancy into
Early Warning dissemination is important, avoiding "singular dependence on one communication device or channel" (IFRC, 2012).

Gauge readers follow pre-defined information flow charts (Shrestha et al., 2014), using mobile phones to call local state and non-state actors (District Disaster Management Committees, the police, the army, Community Disaster Management Committee members, media), to disseminate warnings, and to initiate evacuations (Zurich, 2015). The Chief District Officer simulta-
neously mandates security forces to communicate warnings to police posts, army posts, and local FM radio stations to enable wider community dissemination (Gautam and Phaiju, 2013). Contact details, communications channels and dissemination procedures are prepared in advance (Shrestha et al., 2014). Following initial dissemination of early warning, Local Disaster Management Committee (LDMC) Task Forces are called into action, ensuring all at-risk groups, particularly groups of higher vulnerability, are informed and assisted to respond or evacuate (Shrestha et al., 2014). Dissemination takes place through a
range of appropriate technology channels including sirens, telephone, megaphones, house visits, drums and radio (Gautam and Phaiju, 2013).

### 3.1.4   Response Capability

Response capability focuses on building local to national capacities to respond appropriately to early warning, putting in place well defined response plans whilst building upon local capacities and knowledge (UNISDR, 2005). Actions focus on
strengthening the "capacity of at-risk communities and volunteers to receive, analyse and act-on warnings" (IFRC, 2012).

Awareness-raising programmes utilise a wide range of media and diverse approaches. These may include FM radio, posters, calendars, leaflets, wall paintings, song competitions, street theatre, and school art and essay competitions, to convey key information to support appropriate preparedness and reaction to early warning (Gautam and Phaiju, 2013).

Response capabilities are enhanced through pre-defining response options, roles and responsibilities (including identifying
evacuation routes and safe areas); ensuring teams have access to dissemination and response materials (e.g. loud speakers, life vests, rope), and embedding response plans in wider contingency plans that coordinate across multiple levels from local to national (Mercy Corps and Practical Action, 2010). Individuals designated as 'first receivers' of early warning messages can be trained in interpreting, repackaging and communicating such messages, to ensure they are disseminated in a manner appropriate to the target group (IFRC, 2012).
Response capabilities are also enhanced through practising and testing response plans through mock drills; and undertaking post-event reviews to learn from past hazard events (WMO, 2015). "A community is deemed "response capable" when they know, have practised and have the means to engage in appropriate response actions" (IFRC, 2012).



### 3.1.5 Successes and Limitations

Current CBEWSs in Nepal are expected to give vulnerable communities around two-six hours of preparation time in the event of a flood (Practical Action and Mercy Corps, 2012). The effectiveness of these CBEWSs in Nepal has been demonstrated on a number of occasions including during the 2010 monsoon in Banke district (Mid-Western Nepal), where the district gauge reader

upstream informed downstream communities of rising water levels, enabling the safe evacuation of flood prone communities in Binauna (Practical Action, 2011). Likewise during the 2014 floods in West Nepal, CBEWSs worked effectively in the Karnali and West Rapti basins where communities received flood warnings and were able to respond. Tragically the CBEWS did not succeed in the nearby Babai basin and lives were lost (Zurich, 2015).

Three significant limitations of the current river level monitoring based Early Warning System have been identified. Firstly,

the current system is reliant on real time water level readings, yet both automatic and manual systems are susceptible to failure during extreme rainfall events (as in Babai where the hydro-met station was washed away) (Zurich, 2015). Secondly lead times are short, especially where rivers convey water rapidly, a common situation in mountainous catchments. Thirdly, Nepal has a limited density of hydro-met stations, particularly in remote or hard to access areas, meaning some flood prone areas do not have a corresponding upstream gauge (Ibid).

## 4 Incorporating Forecasts into CBEWS

Hydrological forecasts can be used to address the limited lead time of a river level monitoring based early warning system, hence improving the usefulness of CBEWS. A common approach (Cloke and Pappenberger, 2009) to generating such forecasts would be to cascade meteorological forecasts through a hydrological model resulting in predictions of discharge which could be related to the water level threshold used in issuing warnings. In the case of Nepal such an approach is problematic since

suitable hydrological models do not exist for major river basins (Ibid) and in many cases the performance of precipitation forecasts is inadequate for generating flood warnings (Regmi et al., 2011).

In addressing these problems it is important to consider the sustainability of the solution proposed and, in the spirit of CBEWS, its ability to be maintained and developed with limited ongoing external support or expenditure. Smith et al. (2013) identify two distinct horizons for hydrological forecasts. The first of days or weeks relies on numerical weather prediction and

is useful for strategic activities such as planning recovery activities. The second forecast horizon, often of the order of hours, is that required to communicate urgent warnings to local communities so is of more direct benefit in augmenting CBEWS. To generate forecasts with this horizon is, as demonstrated in Section 5, often possible with only rain gauge data by making use of the natural response time of a catchment.

Regardless of the forecast horizon it has been widely recognised by both scientists and users (Penning-Rowsell et al., 2008;

Frewer et al., 1996) that the uncertainty in the forecasts should be recognised and if possible quantified. Within the context of this work a probabilistic framework is used as described in Sect. 4.1.3. In both cases the assimilation of measured water level data to update the model in real-time can be highly valuable in improving the quality of forecasts for subsequent events.





**Table 1.** Typical data requirements for the DBM methodology to be applied in a catchment (Minimal Data requirements in **bold**) (after Smith et al., 2013) Temperature only required if snow melt is a significant runoff generation mechanism mechanism (Smith et al., 2014)

| Variable | Time step | Time period |
| --- | --- | --- |
| Precipitation | 15 minutes, **1 hour** | **10 significant events**, 5+ years |
| Discharge or Water level | 15 minutes, **1 hour** | **10 significant events including baseflow periods**, 5+ years |
| Temperature | 15 minutes, **diurnal profile** | **10 significant events**, 5+ years |

As in previous works using similar methodologies (see Beven et al., 2011; Leedal et al., 2013; Romanowicz et al., 2008, and the references below) the focus is on forecasting and assimilating water levels. This removes the effects of rating curve uncertainties at gauged sites but clearly such models cannot be constrained by mass balance. This may be advantageous for flood forecasting, where a knowledge of rainfall input data may be limited and rating curves extrapolated well beyond the range of available measurements.

## 4.1 Data Based Mechanistic Modelling

The forecasting methodology developed to compliment CBEWS uses a Data Based Mechanistic (DBM) model (Young, 2002) to predict future water levels at sites where warning levels are defined. Data Based Mechanistic models are parsimonious time series models that can be interpreted in a hydrological fashion often as a non-linear transform of rainfall followed by a linear routing component (see for example Young and Beven, 1994; Young, 2002, 2003; Smith et al., 2013). The philosophy behind DBM modelling is not that of fitting a predefined model, nor that of selecting a statistically optimal (in some predefined sense) model and utilising this as a black box in forecasting. Instead it seeks a model structure and parameters, which while fitted to optimise some criterion and maintain parsimony, must also have a mechanistic interpretation as a representation of the physical system. Section 4.1.1 outlines the adaptations to the model structure and estimation used in this application to ensure this.

While DBM modelling can be used along with forecasts of precipitation to address longer lead times (e.g. Alfieri et al., 2011) they are well suited to shorter forecast horizons since these hydrologically interpretable models can be expressed in a state space form (Sect. 4.1.3) that allows for rapid and robust data assimilation using linear filtering techniques and quantification to the uncertainty in the forecast.

A further advantage of DBM modelling in the context of augmenting CBEWS is the relatively limited data required for their development. Table 1 outlines the typical data requirements for the constructions of a DBM model representing a rainfall-water level relationship. Similar data periods are required for a model representing flood routing driven by an observed upstream water level.

There is no requirement for physiographical information (such as a digital terrain model or land surface characteristics) beyond readily available metadata such as catchment maps showing rain gauges and river connectivity. Nor are further observations such as global radiation or wind speed required





### 4.1.1 Building DBM forecast models

In this section the basis of the DBM models proposed is given. These represent a constraint on the more general model formulations presented in for example Smith et al. (2013) (see also Romanowicz et al., 2006, 2008; Beven et al., 2011, and the references within). An alternative description can be found in Smith (2016) which outlines FloodForT, the software package developed by the authors in the R statistical computing language (Ihaka and Gentleman, 1996) to implement the methodology and provide a convenient web-based Graphical User Interface (GUI).

Consider the estimation of a model of the hydrological response (water level) at a single gauge using a single known input series (typically a catchment average precipitation estimate or an upstream water level/discharge). The foundation of the models used is the linear transfer function. Equation 1 describes a first order discrete time linear transfer function between an output $\mathbf{o} = (o_1, \ldots, o_T)$ and input $\mathbf{u} = (u_1, \ldots, u_T)$, indexed by time ($t$), using the backward shift operator $z^{-1}$ ($z^{-1}o_t = o_{t-1}$) and $d$ time steps of pure, advective time delay. The value of $d$ limits the forecast lead time available from the observed input data.

$$o_t = \frac{\beta}{1 - \alpha z^{-1}} u_{t-d} \tag{1}$$

A first order term of this type represents a non-conservative linear tank whose approximate time constant, representing the decay of the output when there is no input, is given in terms of the data time step $\Delta t$ as $-\Delta t \log(-\alpha)$. The non-conservative nature of the tank is characterised by its Steady State Gain (SSG) given by $\beta/(1 - \alpha)$ which represents the value the output would attain if a unit input was applied indefinitely. Consistency with the belief that the change in output produced directly by $u_{t-d}$ is both positive and decays in time implies that both the time constants and steady state gains take positive values. To ensure this the following parameters restrictions are applied:

$$0 < \alpha < 1 \tag{2}$$

$$0 < \beta \tag{3}$$

Linear tanks have be coupled in many ways to produce representations of flow routing, for example in series to generate the Kalinin-Milyukov-Nash-cascade. While most of these combinations can be captured in an unconstrained DBM analysis in this situation the models considered are limited to those that can be made up from first order linear systems in parallel. This structure of linear tanks has often been used as a step towards finding a mechanistic interpretation in past DBM studies (e.g. Romanowicz et al., 2006, 2008; Leedal et al., 2013).

Expanding the sum of the $n$ first order terms gives

$$\frac{\beta_1}{1 - \alpha_1 z^{-1}} + \ldots + \frac{\beta_n}{1 - \alpha_n z^{-1}} = \frac{b_0 + b_1 z^{-1} + \ldots + b_{n-1} z^{-n+1}}{1 + a_1 z^{-1} + \ldots + a_n z^{-n}} \tag{4}$$

This indicates that the constraints on the general form of DBM model lie both in the model structure as well as its parameterisation.

In the case of $n$ parallel pathways of response their combined output ($\sum_{i=1}^{n} o_{t,i}$) is related to the observed ouput $y_t$ using two additional terms. The first of these $\mu$ is a constant indicative of a baseflow output value while the second $\epsilon_t$ represents a



stochastic noise. The resulting relationship is given by

$$y_t - \mu = \sum_{i=1}^{n} o_{t,i} + \epsilon_t \tag{5}$$

While in limited cases (e.g. Lees et al., 1994) a linear transfer function may adequately capture the dynamics of a system for flood forecasting purposes for forecasting the future response of the system it is more common that a non-linear linear representation of the input-output dynamics is required. Following Young and Beven (1994) non-linearity is introduced by incorporating a gain series $\mathbf{f} = (f_1, \ldots, f_T)$ and altering Eq. 1 so that

$$o_t = \frac{\beta}{1 - \alpha z^{-1}} f_{t-d} u_{t-d} \tag{6}$$

Models of this form, using a number of different methods to calculate $\mathbf{f}$ have proved acceptable representations of a variety of catchments (see for example, Beven et al., 2011; Romanowicz et al., 2006, 2008; Leedal et al., 2013). In the later example two parametric representations of $\mathbf{f}$ are used. The first is the widely used power law non-linearity where $f_t = y_t^{\phi} : 0 < \phi < 1$. Here $y_t$ is used as a surrogate for catchment wetness. The constraints on $\phi$ ensure that $f_t$ is strictly increasing as $y_t$ increases. This is indicative of wetter catchments resulting higher flows. The asymptotic convergence of $f_t$ for increasing $y_t$ produces an effect akin to a catchment becoming saturated. In fact under certain assumptions it can be shown theoretically that the value of $\phi$ in the power law is related to the distribution of the topographic index values utilised for example in TOPMODEL.

The second parametric description of the input non-linearity available is a sigmoid where:

$$f_t = \frac{1}{1 + \exp\left(-\phi_1 \left(y_t - \phi_2\right)\right)} \tag{7}$$

Again $y_t$ is used as a surrogate for catchment wetness and $f_t$ is strictly increasing if $\phi_1 > 0$. The additional flexibility offered by this functional form allow for greater flexibility in the location of the most rapid change in $f_t$. This can be useful in representing catchments where there may be a distinct 'wetting up' phase before a parts of the catchment produce a significant response to the input.

### 4.1.2 Identification and estimation of the Model

Consider the identification and estimation of a hydrological model of the form outlined above. Presuming $\mu$ can be well estimated directly from the values of the observed output (often the minimum value is an adequate estimate) the remaining parameters to be estimated for a model structure given by $n$, $d$ and the chosen non-linearity are $\theta = (\alpha_1, \ldots, \alpha_n, \beta_1, \ldots, \beta_n, \phi)$. A Generalised Method of Moments (GMM, Hall and Inoue, 2003) estimation methodology is used where $\theta$ is chosen to satisfy

$$\mathrm{E}\left[\frac{\partial \epsilon_t}{\partial \theta} \epsilon_t\right] = \mathbf{0} \tag{8}$$

For a linear model with $n = 1$ this is consistent with the Refined Instrumental Variable (RIV) estimation method introduced in Young (1976) (See Young (2011) for a more recent description) widely used within DBM modelling. For a linear model





**Table 2.** Summaries of the model fit expressed in terms of the time indexed observed values $(y_t)$, deterministic model output $(x_t)$ and realisations of the forecast distribution $(F_t)$ denoted $X_t$ and $X_t'$.

| Value | Formula | Description |
|---|---|---|
| $R_t^2$ | $1 - \frac{\sum_t (y_t - x_t)^2}{\sum_t (y_t - \bar{y})^2}$ | Fraction of the variance of the observed data explained. Values approaching 1 are preferred |
| YIC | $\log\left(\sum_i \frac{\Sigma_{i,i}}{\theta_i^2}\right) + \log\left(1 - R_t^2\right)$ | An information criteria the trades of the identifiability of the parameters (first term) with the model performance (second term). On a log scale, more negative values are to be preferred. |
| Bayesian Information Criteria (BIC) | $n\log\left(\sum_t (y_t - x_t)^2\right) + (k+n)\log(n)$ | An information criteria offering an alternative trade off between the number of parameters $k$ and the fit to the $n$ observation. The formula given presumes independent and identically distributed errors. |
| Continuous Ranked Probability Score (CRPS) | $\frac{1}{2}E_{F_t}|X_t - X_t'| + E_{F_t}|X_t - y_t|$ | An assessment of forecast performance on the same scale as the observations which is based on summaries of precision (first term) and accuracy (second term). Corresponds to the Mean Absolute Error it the case of a deterministic forecast. |

with higher $n$ both the RIV and criteria used here can be considered a GMM formulations of a least squares estimation problem though due to the different parameterisations they do not exactly correspond. For non-linear systems the criteria is in keeping with the methods outlines in (Beven et al., 2011) and used in, for example, Leedal et al. (2013). To provide a robust estimate of $\Sigma$ the covariance of the estimate of $\theta$ when the error series $(\epsilon, \ldots, \epsilon_T)$ has unknown, possibly non-stationary and

heteroscedastic, error properties the Heteroscedastic, Auto-covariance Consistent (HAC) estimator of Newey and West (1987).

Since the estimation of the hydrological model is rapid an exhaustive search of the combinations of model non-linearity, $n$ and $d$ is performed. For each estimated model a number of summary statistics (Table 2) are produced. Those which trade off model fit and parsonimity, such as the YIC, can then be screened to remove poor or over parameterised models. Those that remain can undergo a more detailed analysis using for example the thresholded RMSE and visual inspection of the model

hydrographs (particularly focusing on the timing of the rising limbs) both before after the estimation forecast uncertainty. Such an approach allows the consideration of the other requirements since it may be acceptable to trade off the quality of the forecast for increased lead time.



### 4.1.3 Forecast uncertainty and Data Assimilation

Acknowledging the uncertainty in forecasts of future output observations requires that both the sources of the uncertainty and their treatment in the forecasting process be recognised. This offers a starting point for the use of uncertainty representations within the frameworks for flood response. A number of different sources of uncertainty are identified in Smith et al. (2013):

1. uncertainty in observed rainfall fields, only partially captured by interpolation of raingauge data and calibration of radar data;

2. uncertainty in antecedent conditions, only partially captured by running hydrological models continuously;

3. uncertainty in calibration of model parameters, only partially captured by estimation of parameter distributions and their dependence on uncertainty in observations used in model calibration or data assimilation (which may be particularly

high at flood levels);

4. uncertainty in model structures in representing runoff generation and routing processes.

The first three of these sources are addressed in part through the use of noise terms in the data assimilation scheme which reflect the error in the evolution of the model states. and in part through a non-parametric quantification of the forecast errors. The third is also addressed by the implementation of models within a stochastic parameter and error framework. The deductive

method of model selection outlined in Sect. 4.1.2 allows for the exploration of the potential impact of uncertainty in the model structures, although in forecasting on one model structure is utilised.

Assimilating observations of the output to improved the forecasts generated by the DBM models outlined in Sect. 4.1.3 can be considered as a special case of the a more general state space framework for non-linear models (e.g. Liu and Gupta, 2007). The assimilation process used consists of two steps. In the first the probability distribution of the states at time $t+1$ given their

distribution at time $t$ and new pair of observations is derived. The second step evaluates the predictive distribution of $d$ future output observations given the observed inputs, stochastic noise terms and the initial distribution of the model states.

Up to time $t+d$ the input to the model is known since both input and output are observed allowing any gain ($f_t$) to computed. The uncertainty in the initial states of the model therefor corresponds to the uncertainty in the response of each of the first order linear systems. The evolution of these unknown states is governed by linear equations hence, under the assumption

that distribution of the unknown states is symmetric, uni-modal and unbounded, the linear Kalman filter offers an optimal computational scheme for data assimilation and forecasting.

The assumptions of the linear Kalman filter, particularly that of the potential states being unbounded, are inconsistent with the values that would be expected based on a perceptual model. Therefor, as in Smith et al. (2014), a two stage process is used. Firstly the noise variance ratios of the linear filter are selected to minimise the sum, over all lead times and time steps, of the

squared errors of the expected forecast. Following this the expected value of the the forecasts offered by the linear filtering are calibrated to the observed output using quantile regression. This handles the uncertainty arising from all the sources noted earlier and not accounted for by the data assimilation in a lumped fashion. A number of quantile regression techniques have





been used in hydrology (see Weerts et al., 2011, and references within). In this work the quantile regression follows the non-parametric approach outlined in Yu (1999); using initial smoothing based on the nearest 10% of the data points; to predict the difference between the observed output and the expected value of the forecast.

## 5 Application to Karnali Basin

The Karnali basin lies in Western Nepal. Three main tributaries (West Seti, Thuli Bheri and Karnali) drain an area of 45,000 km$^2$ above the Chisapani gauge station (Zurich, 2015) which marks the boundary between the upper catchment dominated by steep sided mountainous valleys and the lower catchment consisting of flood plain which runs to the Indian border (Figure 1). In the Karnali basin, significant floods occurred in 1983, 2009, 2013, and most recently in 2014 (Zurich, 2015). A CBEWS has been successfully operating in the lower catchment since 2010. The CBEWS currently offers between 2 to 6 hours of warning
to those impacted by flooding. It is felt that increasing the lead time would be particularly beneficial in reducing the impacts of flooding for downstream communities by providing them additional time to prepare for floods.

In this demonstration, the water level at Chisapani is forecast using catchment averaged rainfall derived from the three most reliable automated rain gauges (Chisapani, Dipayal and Asaraghat) which have been operational since 2011. Data on an hourly time step is used. As can be seen in Fig. 2 there may be significant missing or poor quality data outside of the monsoon period.
The marked deviation between the rain gauge readings in Fig. 2 suggests a high degree of spatial patterning in the precipitation.

Two experiments are performed which mimic a plausible operational deployment. In the first experiment a forecasting model is estimated using data from 2011 and 2012 then used to predict the 2013 monsoon and flood. This provides a stern test since 2013 contains a flood event whose magnitude, approximately 14 m, exceeds those in the calibration period by around 4 m. In the second experiment data from 2011 to 2013 is used to estimate the model and 2014 flood which peaked at over 16 m is
forecast.

Currently in the Karnali river basin, an anticipatory warning is issued if the water level at the Chisapani gauge rises above 10 m, with a danger warning (indicating the probability of severe flooding downstream) issued at 10.8 m. The focus of these experiments is therefor not just in prediction of the peak magnitudes but also when the forecasts indicated crossing of these warning thresholds.

### 5.1 Experiment 1


The value of $\mu$ was estimated from the calibration data to be 3.07m. Table 3 shows the performance of the a selection of the estimated models ranked by the YIC criteria. No models with three parallel pathways of response or with the sigmoid non-linearity are shown since these return a high YIC value due to the high level of uncertainty in the estimation of the parameters. The similarity of the $R_t^2$ and YIC values indicates that no individual model structure is, in terms of these criteria, superior to
the others. Visual inspection of the model hydrographs suggests that the models with $n = 2$ capture the dynamics of the system markedly better those with $n = 1$. Regardless of the non-linearity models with $d > 6$ appear to have poor correspondence to





**Table 3.** Summaries of the performance of a subset of models during the calibration period (1st Apr. 2011 - 30th Sept.2012)

| Non-linearity | $n$ | $d$ | $R_t^2$ | MAE | BIC | YIC |
|---|---|---|---|---|---|---|
| None | 2 | 8 | 0.802 | 0.506 | 34412.30 | -17.267 |
| None | 2 | 8 | 0.802 | 0.506 | 34412.30 | -17.267 |
| None | 2 | 7 | 0.802 | 0.622 | 34392.15 | -17.212 |
| None | 2 | 6 | 0.802 | 0.620 | 34375.27 | -17.134 |
| None | 2 | 5 | 0.803 | 0.616 | 34355.40 | -17.079 |
| None | 2 | 4 | 0.803 | 0.613 | 34342.57 | -16.987 |
| None | 2 | 3 | 0.803 | 0.610 | 34327.32 | -16.917 |
| None | 2 | 2 | 0.803 | 0.606 | 34314.13 | -16.835 |
| None | 2 | 1 | 0.804 | 0.603 | 34298.01 | -16.771 |
| Power Law | 2 | 9 | 0.828 | 0.481 | 32368.67 | -16.244 |
| Power Law | 2 | 8 | 0.828 | 0.480 | 32313.62 | -16.181 |
| Power Law | 2 | 7 | 0.829 | 0.480 | 32258.15 | -16.124 |
| Power Law | 2 | 6 | 0.830 | 0.480 | 32205.95 | -16.035 |
| None | 1 | 2 | 0.800 | 0.499 | 34536.61 | -16.017 |
| None | 1 | 3 | 0.800 | 0.499 | 34553.34 | -16.003 |
| None | 1 | 5 | 0.799 | 0.500 | 34587.50 | -15.995 |
| None | 1 | 8 | 0.798 | 0.500 | 34645.20 | -15.995 |
| None | 1 | 7 | 0.799 | 0.500 | 34625.23 | -15.994 |
| None | 1 | 6 | 0.799 | 0.500 | 34606.38 | -15.991 |
| None | 1 | 9 | 0.798 | 0.500 | 34665.07 | -15.990 |
| None | 1 | 1 | 0.800 | 0.499 | 34518.39 | -15.989 |
| None | 1 | 4 | 0.800 | 0.500 | 34570.68 | -15.989 |
| Power Law | 2 | 5 | 0.830 | 0.479 | 32151.79 | -15.963 |
| Power Law | 2 | 4 | 0.831 | 0.479 | 32103.44 | -15.854 |
| Power Law | 2 | 3 | 0.831 | 0.478 | 32054.01 | -15.775 |
| Power Law | 2 | 2 | 0.832 | 0.478 | 32007.97 | -15.669 |
| Power Law | 2 | 1 | 0.833 | 0.478 | 31960.19 | -15.557 |
| Power Law | 1 | 9 | 0.815 | 0.501 | 33382.66 | -13.458 |
| Power Law | 1 | 8 | 0.816 | 0.500 | 33344.82 | -13.451 |
| Power Law | 1 | 7 | 0.816 | 0.500 | 33307.00 | -13.440 |
| Power Law | 1 | 6 | 0.817 | 0.500 | 33269.80 | -13.429 |
| Power Law | 1 | 5 | 0.817 | 0.499 | 33234.05 | -13.422 |



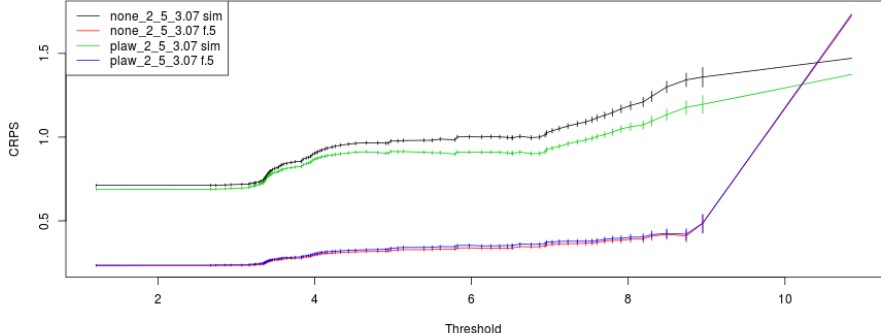

**Figure 3.** CRPS of the two pathway model with a 5h time delay with (plaw_3_5_3.07) and without (none_2_5_3.07) the power law non-linearity. Results are shown both for the estimated model (sim) and for the 5h lead time forecasts (f.5). Error bars represent +/- 2 standard deviations of the mean of the CRPS values of the time steps whose observed value exceeds the threshold.

the rising limb of the hydrograph. Given this a time delay of 5h ($d = 5$) was chosen to maximise the forecast lead time while retaining an acceptable fit.

Figure 3 shows the marked improvement that results from the use of data assimilation and the uncertainty representation. After data assimilation has been applied performance between the two models shown is very similar, however due to its superior
performance without data assimilation the model with the power law non-linearity is to be preferred.

With only a single flood event in the validation period of the 1st June to 30th Sept. 2013 detailed analysis of models performances in forecasting the crossing of the warning levels is not possible. However Fig. 4 issued at 16:00 17th June 2013 shows that the model discriminates between between exceedance and non exceedance of the warning and danger thresholds even at its maximum lead time.

**5.2  Experiment 2**

The analysis performed in Experiment 1 was repeated using a longer calibration period, the 1st Apr. 2011 - 1st Sept.2013. This contains data for a further monsoon period during which a flood event occurred. The impact of this on model estimation can be clearly seen in the results shown in Tbl. 4. In contrast to Tbl. 3 there is a clear preference for model with $n = 2$ and some preference for a power law non-linearity.
Taking, as in Experiment 1, the model with a power law non-linearity, $n = 2$ and $d = 5$ (but revised parameter estimates) Figure 5 shows the forecasts issued at 23:00 14th August 2014. Again the model clearly discriminates on the crossing of the warning thresholds.



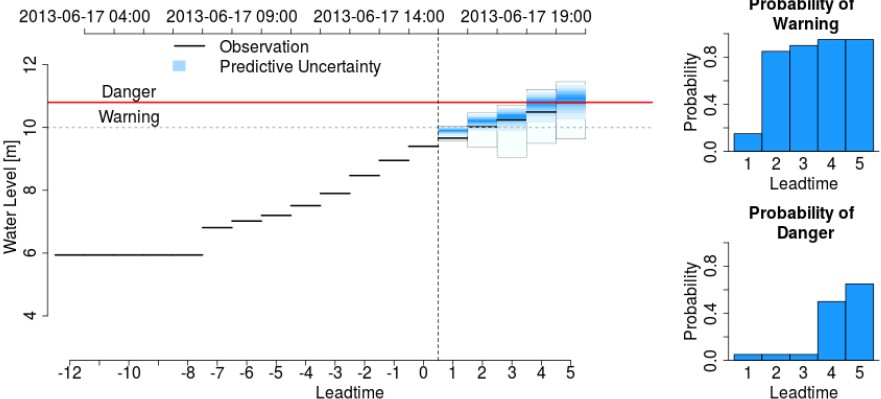

**Figure 4.** Forecast issued at 16:00 17th June 2013 showing on the left the predictions of future water level (blue) and observed data (black). On the right the probability of crossing the respective warning levels at each lead time is shown.

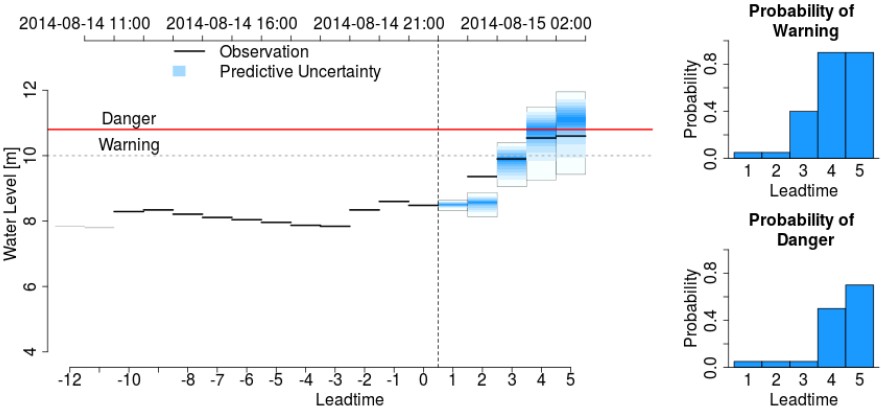

**Figure 5.** Forecast issued at 23:00 14th August 2014 showing on the left the predictions of future water level (blue) and observed data (black). On the right the probability of crossing the respective warning levels at each lead time is shown.

## 6 Conclusions

This paper has outlined the current status of CBEWSs in Nepal, the benefits they have offered to communities vulnerable to flooding, as well as some ongoing challenges. It has presented a simple and robust probabilistic forecasting methodology for extending warning lead times. The Nepal Department of Hydrology and Meteorology, working alongside other partners including Practical Action, has integrated this approach with its real-time data systems to allow operational testing over the 2016 monsoon season.

The case study presented indicates that for the CBEWS operating in the Karnali basin the lead times at which warnings are issued could potentially be increased from the current 2-6 hours to up to 7-11 hours. Achieving these increases will require



**Table 4.** Summaries of the performance of a subset of models during the calibration period (1st Apr. 2011 - 1st Sept.2013). This shows a clearer definition of the preferred model then results in Tbl. 3

| Non-linearity | $n$ | $d$ | $R_t^2$ | MAE | BIC | YIC |
|---|---|---|---|---|---|---|
| Power Law | 2 | 2 | 0.807 | 0.549 | 39697.05 | -22.236 |
| Power Law | 2 | 3 | 0.806 | 0.550 | 39804.56 | -22.202 |
| Power Law | 2 | 1 | 0.808 | 0.548 | 39596.65 | -22.176 |
| Power Law | 2 | 8 | 0.797 | 0.557 | 40504.23 | -22.161 |
| Power Law | 2 | 9 | 0.796 | 0.558 | 40655.08 | -22.161 |
| Power Law | 2 | 5 | 0.803 | 0.553 | 40056.37 | -22.134 |
| Power Law | 2 | 4 | 0.804 | 0.551 | 39926.08 | -22.133 |
| Power Law | 2 | 7 | 0.799 | 0.555 | 40349.75 | -22.114 |
| Power Law | 2 | 6 | 0.801 | 0.554 | 40203.86 | -22.102 |
| None | 2 | 9 | 0.794 | 0.547 | 40796.81 | -21.719 |
| None | 2 | 8 | 0.795 | 0.547 | 40719.24 | -21.591 |
| None | 2 | 7 | 0.796 | 0.547 | 40639.45 | -21.450 |
| None | 2 | 6 | 0.797 | 0.547 | 40571.15 | -21.306 |
| None | 2 | 5 | 0.798 | 0.547 | 40492.91 | -21.167 |
| None | 2 | 4 | 0.798 | 0.547 | 40437.03 | -20.998 |
| None | 2 | 3 | 0.799 | 0.547 | 40382.34 | -20.855 |
| None | 2 | 2 | 0.799 | 0.546 | 40339.38 | -20.709 |
| None | 2 | 1 | 0.800 | 0.546 | 40298.90 | -20.502 |
| None | 1 | 6 | 0.779 | 0.541 | 41894.44 | -16.979 |
| None | 1 | 5 | 0.780 | 0.541 | 41833.68 | -16.975 |
| None | 1 | 7 | 0.779 | 0.541 | 41954.98 | -16.969 |
| None | 1 | 4 | 0.781 | 0.540 | 41776.89 | -16.964 |
| None | 1 | 8 | 0.778 | 0.542 | 42017.90 | -16.963 |
| None | 1 | 9 | 0.777 | 0.542 | 42080.51 | -16.960 |
| None | 1 | 3 | 0.782 | 0.540 | 41720.96 | -16.959 |

connections between the forecasts and other components of the CBEWS to be made. Ongoing work is focused both on this integration and assessing the ability of the forecasting methodology to increases the warning lead times in other basins.

*Author contributions.* PJS undertook the modelling and coordination of the manuscript. SB provided sections on CBEWSs. SD provided background information about Nepal and details of the development of CBEWs there.



*Acknowledgements.* PJS has been supported by the knowledge exchange placement funded by the NERC PURE initiative. Funding from the Zurich Flood Resilience Alliance has also supported the work presented.



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
