# Peer review of "Community Based Early Warning Systems for flood risk mitigation in Nepal"

_Natural Hazards and Earth System Sciences, 2016_

## Referee Comment (RC1) · EM Stephens (Referee) · 12 May 2016

This is an interesting paper in the application of probabilistic forecasting for flood early warning in Nepal. There is an existing community-based early warning system in the area, using an upstream gauge observation to provide an early warning of imminent flooding, but the need for a longer lead time forecast motivates the authors to pursue a probabilistic model-based approach.

This work has clear merit and is within the focus of the journal, though I would ask the authors to improve on how the two distinct sections (development of community-based early warning systems, and development of the probabilistic flood forecast) link together, especially given how moving from a deterministic to probabilistic approach will clearly influence how the early warning system is set-up.

[Figure]

Recommendations:

1) Sections 1&2 are quite general. I believe it would be beneficial to the paper for these sections to be more focussed on the case-study catchment, situating this information within the wider context of resilience building and flood hazard in Nepal where relevant.

2) Figure 1 is also quite general – I recommend editing it to highlight the case-study catchment, along with the location of the rain and river gauges. I'm not sure whether it is deliberate or just due to NHESS formatting, but the Figures should be included as close to where they are relevant as possible.

3) Table 1 includes information on minimum data requirements for the application of the methodology. It would be interesting in Sections 3.1.1-3.1.3 if there could be some discussion of the minimum requirements for each of these components for establishing a community-based early warning system.

4) Depending on the target audience of this paper, the flow of the text could perhaps be improved by moving Section 4.1 to a Supplementary Information file, and providing a summary for a lay-person. The important message for a practitioner is perhaps that this model is available via an R package and GUI, and can be applied to anywhere where there is gauged data? In some ways having an overly technical explanation might hinder the uptake of the model for future applications!

5) For Table 1, are these the minimum data requirements for the calibration period of the model, or for both the calibration and evaluation periods?

6) I feel that there is a section missing on how these newly-developed probabilistic forecasts can be applied. The forecast provides the probability of exceeding a warning threshold. Section 3.1 describes aspects such as dissemination and communication, and response capability. How has the CBEWS had to be adapted to enable decisions to be made from probabilistic forecasts? Has the community received training in probabilistic forecasts, or have procedures been put in place so that the community can

follow them without needing to interpret the probability themselves? Answering these kind of questions would be really valuable to the academic and practitioner communities working on Early Warning Systems.

7) The conclusion would benefit from some additional discussion on the wider context of the work carried out.What are the key messages for the development of probabilistic forecasts / CBEWS elsewhere? Could the methodology be easily applied elsewhere? Would it achieve the same increase in warning lead-time? (Presumably dependent on catchment size and driver of flood?)

Specific Comments

P1L14: I think in this context you mean 'populus' (n) not 'populous' (adj)

P1L24: Regional not region

P2L5: At this point it would be helpful to define what a Community Based Early Warning System is.

Fig2: It would be helpful if Fig1 showed where the gauges are located.

P4L27: Could the map in Fig1 also show the location of the other Practical Action CBEWS projects?

P5L3: Could there be a brief description of what a Community Risk Assessment involves? Who leads these assessments? How are they carried out?

3.1.3 to 3.1.4: This is touched upon slightly; there are clear procedures to follow, what is done to ensure that these procedures are followed? Is anyone held accountable if they are not?

P7L8: Why did it fail? (and later, perhaps discuss/ comment on if the model-based methodology would have continued to work in this situation)

P13L19: 'the' 2014 flood

P13L23: therefore

P13L26: either 'the' or 'a' selection

---

## Referee Comment (RC2) · E. Coughlan de Perez (Referee) · 25 May 2016

Overall, the article makes a useful contribution to the need for innovative flood modeling approaches in data-scarce areas. The concepts outlined here could be applied elsewhere, and many of my comments suggest that the authors provide more explanation for the reader to understand the applicability to other situations.

Comments on the Early Warning Systems section

In general, it is nice to have an overview of the state of early warning systems in Nepal, and the variety of options and methods that have been trialed.

As a reader new to this piece, I had a lot of questions when trying to follow the description – specific questions are below, but I suggest to go through section 3 to read

for clarity. The link between this section and the proposed EWS is not entirely clear; it would be good to shorten this section and focus on the main points and problem statements. Here are some specific questions and points of confusion when reading:

Page 2 line 5: What is a "Community Based Early Warning System"? What qualifies something as a CBEWS? You give a slightly longer explanation on page 4, but to the reader, it would seem that a normal EWS also helps communities prepare for and respond to hazard events, which is the description you provide for a CBEWS.

Page 3: What is the axis in this plot? Where are these gauge stations located?

Page 4 section 3.1: Can you explain further? What do these CBEWS entail? It is not really clear if what you go on to describe (e.g. 3.1.3) is an explanation of these CBEWS or just an explanation of the national system.

Page 5 section 3.1.1: What are warning and danger levels? Can you explain further what these mean and how they were determined?

Page 5 line 20: How do the communities report the monitoring of river levels to the DHM? Do they receive formal training for this?

Page 6 line 3: Are the forecasts only based on observed water levels upstream, or do they incorporate rainfall?

Page 6 line 21: What are the actions and messages that are being disseminated through these channels? Is it about evacuation only?

Page 8 line 8: Are you predicting future water levels at the upstream gauges (where warning levels are defined) or at downstream locations that are likely to be impacted?

Page 13 line 10: What are the additional actions that could be taken, and how much lead-time is needed for these actions? This is a key point that is missing to make the link between the first section on CBEWS and the modeling endeavor described second.

You mention that there have been thousands of flooding events in the past 40 years in Nepal. How many of these were anticipated by forecasts? What kind of action was taken to prepare for these floods?

Comments on the proposed EWS

This section offers a novel application of a flood modeling system in places with little data. However, it is not clear to the reader whether the short time period available is a good enough training period to accurately represent uncertainties going forward. How could this be ascertained? Also, why is a lead-time of 5 hours chosen? How would the skill of the model change at longer lead-times?

These questions could be answered simply by expanding the conclusion, in order to summarize the pros and cons of this approach and the situations in which this would be most relevant. From the point of view of someone in another catchment interested to replicate this approach, how do the data requirements for this system compare to those of other options, and how does the processing requirements of this proposed system compare to other hydrological modeling choices? Also in the conclusion, it would be of interest to the reader to learn more about how this has been integrated with the CEWS that were described in the earlier section, and what type of results are anticipated from the testing of the system.

Some specific questions and comments:

Page 12: There are a number of other uncertainties when it comes to using this information for an early warning system. For example, the uncertainty in whether the danger level corresponds to actual impact (e.g. if a village moves or if agricultural patterns change).

Page 12: Paragraph starting on line 22 is difficult to follow, perhaps also because of some spelling/grammar errors.

Page 13: Paragraph starting on line 21: If the warning is issued using observations at

Interactive
comment

[Figure]

Chispani gague, how long does it take for the floodwaters to arrive downstream? In the new system, are you forecasting the level of Chispani in order to give lead-time to that community, or forecasting the level at Chispani in order to give extra lead-time to the people downstream? In particular, it sounds like you are offering an additional lead time of only 5 hours, correct? It would be good to explain further all of these details.

Page 14: Figure 3 is not easy to understand. What do you mean when you say "values of the time steps whose observed value exceeds the threshold"?

Page 15: You demonstrate that the model would have accurately foreseen the crossing of the warning/danger levels five hours before the floods of 2013 and 2014. However, are there any other instances in the model hindcasts that would have unnecessarily crossed the danger level and given a false alarm? What is the probability of a false alarm?

Page 15: How frequently do your forecast cross the danger level? How does this compare to the frequency of the danger level happening in real life?

Page 15: In general, the accuracy of the model for low flows is not of particular interest in this case, as the goal is to provide early warnings for extreme floods. It would be of interest to the reader to have more statistics on the extreme events. What data is available for you to work with? Is it possible to create hindcasts of your model? If so, can you calculate the extent to which these forecasts would match up with the historical records from Desinventar?

Table 3 and Table 4 provide summary of the model performance during the calibration period, but it would also be good for the reader to see how each model performed during the test period (non-calibration period). How were the calibration periods and test periods selected?

Page 16 Line 5: Which non-linearity is being used, and why? How are they actually testing this? It would be of interest to include more details on this.

---

## Author Comment (AC1) · 20 Jul 2016

We (the authors) thank EM Stephens for her review of the paper and for the constructive suggestions. In the following response the response to the reviewers comments is given in *italics* for clarity.

This work has clear merit and is within the focus of the journal, though I would ask the authors to improve on how the two distinct sections (development of community-based early warning systems, and development of the probabilistic flood forecast) link together, especially given how moving from a deterministic to probabilistic approach will clearly influence how the early warning system is set-up.
*The authors acknowledge that in the submitted manuscript there may appear a disconnect between the two sections. We believe that revisions to the text (particularly in*

*response to Recommendations 1, 4 & 6) can be used to address this and make the paper more appealing to a broader audience.*

**Recommendations**

1. Sections 1 & 2 are quite general. I believe it would be beneficial to the paper for these sections to be more focussed on the case-study catchment, situating this information within the wider context of resilience building and flood hazard in Nepal where relevant.

   *The more general nature of these sections was commented on positively by the second reviewer. However the authors agree that it may be constructive to introduce the case study earlier in the paper to focus the discussion if this could be done without losing the overview currently offered.*

   *Perhaps, we can use the Introduction section to build upon what we have already described and put an emphasis on Karnali basin in particular linking it to the revised Figure that highlights the real time rainfall and water level sensors as mentioned by Recommendation 2*

2. Figure 1 is also quite general – I recommend editing it to highlight the case-study catchment, along with the location of the rain and river gauges. I'm not sure whether it is deliberate or just due to NHESS formatting, but the Figures should be included as close to where they are relevant as possible.

   *We concur that this figure could provide the requested detail of the study catchment. While we will attempt to alter the figure positioning we consider this a production issue for the journal.*

3. Table 1 includes information on minimum data requirements for the application of the methodology. It would be interesting in Sections 3.1.1-3.1.3 if there could be

some discussion of the minimum requirements for each of these components for establishing a community-based early warning system.

*This information could be included (perhaps through an expansion of Table 1 referenced in Section 3) and would make a significant contribution to improving the paper.*

4. Depending on the target audience of this paper, the flow of the text could perhaps be improved by moving Section 4.1 to a Supplementary Information file, and providing a summary for a lay-person. The important message for a practitioner is perhaps that this model is available via an R package and GUI, and can be applied to anywhere where there is gauged data? In some ways having an overly technical explanation might hinder the uptake of the model for future applications!

*Section 4.1 has tried to strike a balance between providing adequate details of the methodology for the user to understand the forecasting methodology (and hence interpret its use, data requirements and application) while not being overly technical. In light of the reviewers comments this balance can be revised and the key messages (as correctly identified by the reviewer) highlighted. A more detailed description could be placed in the supplementary information or left for the reader to find in references provided.*

5. For Table 1, are these the minimum data requirements for the calibration period of the model, or for both the calibration and evaluation periods?

*These are the minimum requirements that have been found to be useful both in practice and in offline experiments. When the amount of data available is at or close to these amounts, the authors would not suggest a 'fixed' division into calibration and evaluation periods as often used in hydrology. Instead the authors have found it more useful to use a cross validation approach. In this, the performance of the model for each event is evaluated using parameters estimated from the remainder of the data. This would be expanded upon in the text.*
[Figure]

6. I feel that there is a section missing on how these newly-developed probabilistic forecasts can be applied. The forecast provides the probability of exceeding a warning threshold. Section 3.1 describes aspects such as dissemination and communication, and response capability. How has the CBEWS had to be adapted to enable decisions to be made from probabilistic forecasts? Has the community received training in probabilistic forecasts, or have procedures been put in place so that the community can follow them without needing to interpret the probability themselves? Answering these kind of questions would be really valuable to the academic and practitioner communities working on Early Warning Systems.

*We concur that a more focused discussion of these issues is required in the paper. However any such discussion should be preceded by acknowledgment of the fact that these procedures are evolving. For the 2016 monsoon, the Nepal Department of Hydrology and Meteorology (DHM) are testing a 'top down' communication strategy, whereby the DHM Central Office would provide advisories to its Basin Offices, which in turn would be passed to district stakeholders including District Emergency Operation Centers (DEOC), security forces and Community Groups. These are well established communication channels on services manned 24/7 during the monsoon season.*

*The probabilistic forecasts provide information regarding the water levels for the next 5 hours and the specifically the probability of water levels hitting identified danger and warning levels. For now, these forecasts would provide enhanced early awareness so that communities downstream and government stakeholders can be primed to remain alert regarding an incoming flood in the next few hours.*

*It remains to be seen if such an approach to disseminating the forecast information is successful and it is hoped to attempt a more direct approach to communicating the forecasts to communities in future years, based on the experience from the pilot in Monsoon 2016. We will provide this context in the paper.*

7. The conclusion would benefit from some additional discussion on the wider context of the work carried out. What are the key messages for the development of probabilistic forecasts / CBEWS elsewhere? Could the methodology be easily applied elsewhere? Would it achieve the same increase in warning lead-time? (Presumably dependent on catchment size and driver of flood?)

*The authors are happy to add such discussion. The methodology used is quite general but is reliant upon community engagement and relevant real time data (as discussed elsewhere in the paper) to allow for the forecasting. Currently models are being developed for both the major (large) river basins of Nepal such as the Koshi and Narayani as well as for smaller basins such as Babai, West Rapti and Kankai to augment existing Community based early Warning Systems. The lead time available will vary by locations and is dependent upon many factors including the dominant channel and geophysical characteristics and spatial distribution of the gauged sites.*

**Specific Comments**

*A number of textual corrections to spelling and grammar highlighted by the reviewer are accepted and do not feature below.*

P2L5: At this point it would be helpful to define what a Community Based Early Warning System is.
*This would be included in the response to Recommendation 1*

Fig2: It would be helpful if Fig1 showed where the gauges are located. P4L27: Could the map in Fig1 also show the location of the other Practical Action CBEWS projects?
*We concur, see the response to Recommendation 2*

P5L3: Could there be a brief description of what a Community Risk Assessment involves? Who leads these assessments? How are they carried out?

*Yes this can be added. Community Risk Assessments are a form of Participatory Vulnerability and Capacity Assessment (PVCA) - participatory risk assessment and mapping processes involving the communities.*

3.1.3 to 3.1.4: This is touched upon slightly; there are clear procedures to follow, what is done to ensure that these procedures are followed? Is anyone held accountable if they are not?

*This can be addressed in a revised Section 3 with more focus on the target location as described above.*

P7L8: Why did it fail? (and later, perhaps discuss/ comment on if the model-based methodology would have continued to work in this situation)

*The failure of the CBEWS in Babai was primarily due to the water level gauge station used for triggering alerts being washed away during high floods. Due to this, communities and stakeholders downstream were deprived of critical information regarding when water levels crossing warning and danger levels. Compounding this, the gauge reader was unable to access the gauge station to provide a manual assessment of the flows ( he was trapped between torrents of water in the trail leading to the station) and damaged his cell phone leaving him unable to contact communities, security forces and district authorities in a timely manner. This led to delays in response and rescue operations which contributed to more than 20 people losing their lives. A further contributing factor was that floodwaters entered places that were deemed safe in past risk mapping exercises and caught communities by surprise.*

*In the situation where data from the gauge whose water levels are being forecast is lost the model will continue to be evaluated and forecasts generated, but without the benefit of data assimilation. This will typically result in a decrease in the forecast quality. The model will however fail to deliver forecasts if the gauge(s) providing the input data (typically precipitation) fail to provide data.*

---

## Author Comment (AC2) · 20 Jul 2016

We (the authors) thank E. Coughlan de Perez for her review of the paper and for the constructive suggestions. In the following pages the response to the reviewers comments is given in *italics* for clarity. Where these responses are factual we would aim to include this information in a revised manuscript.

Overall, the article makes a useful contribution to the need for innovative flood modeling approaches in data-scarce areas. The concepts outlined here could be applied elsewhere, and many of my comments suggest that the authors provide more explanation for the reader to understand the applicability to other situations.
*We would be happy to improve the paper with this aim.*

Comments on the Early Warning Systems section In general, it is nice to have an

overview of the state of early warning systems in Nepal, and the variety of options and methods that have been trialed. As a reader new to this piece, I had a lot of questions when trying to follow the description – specific questions are below, but I suggest to go through section 3 to read for clarity.
*This suggestion is accepted.*

The link between this section and the proposed EWS is not entirely clear;
*The lack of clarity in the link between the two sections is also highlighted by the other reviewer. We believe the additional section the other reviewer proposes (RC1: Recommendation 6) will address this issue.*

it would be good to shorten this section and focus on the main points and problem statements. Here are some specific questions and points of confusion when reading:

Page 2 line 5: What is a "Community Based Early Warning System"? What qualifies something as a CBEWS? You give a slightly longer explanation on page 4, but to the reader, it would seem that a normal EWS also helps communities prepare for and respond to hazard events, which is the description you provide for a CBEWS.
*A CBEWS is designed with communities as active participants in the EWS design, monitoring and management, not just passive recipients of warnings. Details can be added to this section.*

Page 3: What is the axis in this plot? Where are these gauge stations located?
*A better quality figure will be produced to address this issue. Gauges will be marked on Fig 1*

Page 4 section 3.1: Can you explain further? What do these CBEWS entail? It is not really clear if what you go on to describe (e.g. 3.1.3) is an explanation of these CBEWS or just an explanation of the national system.
*CBEWS were established in Nepal in 2002, expanding across 8 river basins over the next ten years. These CBEWS have over the past 5 years been aligned with/integrated with a national EWS whilst maintaining community driven/managed features. De-*

*tails/clarity can be added to this section.*

Page 5 section 3.1.1: What are warning and danger levels? Can you explain further what these mean and how they were determined?
*The Danger and Warning Levels are the water levels at the gauge used to trigger responses from the CBEWS (typically upstream of the at risk areas) which are believed to correspond to: Warning Level – Bank Full Condition at the site at risk Danger Level – When water over tops the banks, and water enters the communities These are specific to the community at risk and are a based on a mixture of flood hazard mapping (carried out by DHM) and community experience.*

Page 5 line 20: How do the communities report the monitoring of river levels to the DHM? Do they receive formal training for this?
*River gauges in Nepal are maintained by the Government via Part Time Staff who are in nearly all cases members of the local community who are trained to record and monitor river water levels. This results in a situation where, although the government pays for the gauges, information on water levels (in the period immediately leading up to flooding) is provided directly to the local community by one of its members.*

Page 6 line 3: Are the forecasts only based on observed water levels upstream, or do they incorporate rainfall?
*The operational early warning system currently in place in the area usually triggers responses based upon observed upstream river levels. However there have been a few occasions when responses have been triggered by exceeding empirically defined intensity-duration thresholds for precipitation.*

Page 6 line 21: What are the actions and messages that are being disseminated through these channels? Is it about evacuation only?
*Through these channels, information regarding staying prepared for floods, mobile numbers of upstream gauge readers, actions to be taken during floods are disseminated as well as evacuation*

Page 8 line 8: Are you predicting future water levels at the upstream gauges (where warning levels are defined) or at downstream locations that are likely to be impacted? *The predictions are for water level at specific gauges where thresholds have been defined for triggering a CBEWS response. These are typically upstream of the affected areas. The upstream warning levels are calculated to correspond to downstream flood scenarios as described above.*

Page 13 line 10: What are the additional actions that could be taken, and how much lead-time is needed for these actions? This is a key point that is missing to make the link between the first section on CBEWS and the modeling endeavor described second.
*The current very limited lead time warnings times are sufficient for saving lives, but increasing the lead time through modelling offers a number of potential benefits:*

- *Extended lead times allows for a more robust system, currently any failure in communication results in a delay which may render the warning useless.*

- *Current lead-times are insufficient for saving moveable assets, livestock, livelihood tools etc. Additional warning time could help protect livelihoods as well as lives.*

- *Adding to the warning lead time would increase the confidence in this system reaching all at risk groups.*

- *A 2-3 hour lead time for evacuation is especially challenging for vulnerable groups eg disabled, pregnant women, elderly, children – increasing the lead time enables slower and safer evacuation.*

*The last significant flood event (2014) released a warning at eg 1am, at a time when most people were asleep, and evacuation needed to happen at eg 3-4 am. Increasing the lead time gives more likelihood of warning messages and or evacuation occurring*

[Figure]

*during hours when communities are awake/during daylight. Having that same warning available 5 hours earlier at eg 8pm, would enable safer and easier evacuation.*

You mention that there have been thousands of flooding events in the past 40 years in Nepal. How many of these were anticipated by forecasts? What kind of action was taken to prepare for these floods?
*The first CBEWS started in 2002 in East Rapti river basin, Central Nepal. Prior to that there were no operational forecasting and early warning systems in place in Nepal and therefore no coordinated preparation and anticipatory response.*

Comments on the proposed EWS This section offers a novel application of a flood modeling system in places with little data. However, it is not clear to the reader whether the short time period available is a good enough training period to accurately represent uncertainties going forward. How could this be ascertained?
*In practice there is no guarantee that any model will achieve this due to the uniqueness of place and process in hydrology. Best current practice is to (a) build a model that you have some faith will capture the catchment dynamics in new situations (b) test this model using split sample methodologies (i.e. calibration on a period of data and validation on a second period of data). The method proposed makes use of both (a) and (b). It should be noted that the best method of split sample testing will depend upon the quantity of data available, this can be discussed further in the text. The minimum data requirements are based upon experience elsewhere with this type of model and experimentation using random samples of longer data sets*

Also, why is a lead-time of 5 hours chosen?
*Typically in all but the smallest catchments there is a time delay between the rainfall and initial rise in water level at the outlet. This time delay is estimated within the model identification step (Section 4.1.2 & Table 3) where it is referred to as the advective time delay of the model. In this case a value of 5 hours was identified using the methodology outlined. This means that forecasts of the observed output variable (water level) of up to 5 hours lead time can be issued using observed values of the input variable*

(precipitation). The role of the advective delay component of the model will be outlined more clearly in the paper text.

How would the skill of the model change at longer lead-times?
*As outlined in the response above and in the paper (Sec 4.1.1, Pg 8 line 24) forecasts with up to 5 hours lead time can be generated without precipitation forecasts. There are two options for generating forecasts with longer lead times. The first is the use of precipitation forecasts. These are currently unavailable for Nepal at a suitable resolution for modelling at an hourly time step (but see references for examples of their use elsewhere). An alternative approach is to use a time delay (as outlined above) which is longer than that identified as optimal. Selection of such a time delay typically gives a poorer model of the initial response to the input (precipitation) but may be adequate for the capturing the timing of crossing alert levels.*

These questions could be answered simply by expanding the conclusion, in order to summarize the pros and cons of this approach and the situations in which this would be most relevant.
*The other reviewer also noted the need to expand the conclusion. We accept this recommendation and will update accordingly.*

From the point of view of someone in another catchment interested to replicate this approach, how do the data requirements for this system compare to those of other options, and how does the processing requirements of this proposed system compare to other hydrological modeling choices?
*Typically the amount of data needed for the methodology proposed is far less then that used by most other methods. Like most hydrological approaches there is the need for well defined and consistent data but unlike most approaches the methods outlined can make direct use of water level data. This removes the need to maintain a high quality stage-discharge relationship for determining the discharge during flood events from the observed water level. Also the framework is particularly flexible. It can be used either as a lumped rainfall-runoff model (as outlined in the example) or for mod-*

*elling river routing. This means that by learning one modelling approach (and associated software) practitioners can generate forecasting models for either situation.. Also since the model structure is identified from the data this eliminates unnecessary model components resulting in robust estimation with few of the problems of parameter equifinality present in other approaches. Thus unlike many lumped or distributed modelling strategies there is no need for detailed topographic, land use or soil property data to constrain the estimation. Published modelling results for other catchments suggest that even when more detailed models can be constructed the forecast performance of the methodology outlined is often as good if not better at the sites being forecast.*

*The resources required for this type of modelling outlined in the paper are low. Practical exercises given as part of training courses suggest that the modelling approach can be successfully taught over one to two days. Using the software tools developed by the authors, models can be constructed in a few hours on a standard laptop. In an operational setting each new forecast can be generated in around 5 seconds (even on older single core PCs), this includes data extraction and generation of plots of the type shown. Past work has shown that the online generation of forecasts can be practical even on older smart phones.*

Also in the conclusion, it would be of interest to the reader to learn more about how this has been integrated with the CEWS that were described in the earlier section, and what type of results are anticipated from the testing of the system.
*This was raised by the other reviewer as well. It is proposed to add a further section to the paper outlining in more detail how the system has been tested, initial results and potential future developments. This will aid in bringing the two sections of the paper together since many of the challenges and developments lie in the integration of the CBEWS and forecasting methods.*

Some specific questions and comments:

Page 12: There are a number of other uncertainties when it comes to using this in

formation for an early warning system. For example, the uncertainty in whether the danger level corresponds to actual impact (e.g. if a village moves or if agricultural patterns change).

*We agree this is the case. In fact the situation is worse since in the Karnali the area at risk is a low lying inland delta consisting of wide shallow naturally mobile channels with significant sediment movement and braiding which are subject to time varying human impact (irrigation channels being cut, banks eroded due to vehicular river crossing at low flows etc). In such an environment the accurate spatial prediction of impacts is (at least) very challenging and subject to significant uncertainties. Key to the success of the CBEWS is the recognition of this and an acceptance that community input is required in both the initial setting and the ongoing revision of the levels used to trigger alerts and warnings.*

Page 12: Paragraph starting on line 22 is difficult to follow, perhaps also because of some spelling/grammar errors.
*This will be revised.*

Page 13: Paragraph starting on line 21: If the warning is issued using observations at Chispani gague, how long does it take for the floodwaters to arrive downstream? In the new system, are you forecasting the level of Chispani in order to give lead-time to that community, or forecasting the level at Chispani in order to give extra lead-time to the people downstream? In particular, it sounds like you are offering an additional lead time of only 5 hours, correct? It would be good to explain further all of these details.

*From Chisapani, it takes 2-3 hours for flood waters to arrive at the first significant communities approximately 30 km downstream. The settlement at Chisapani itself is mostly set well above the river and has a low flood risk. In the new system we are forecasting the level of Chisapani in order to give an improved lead-time to downstream communities. A forecast with 5 hours lead time at Chisapani results in 7-8 hours lead time at the most affected downstream communities. While this extension in lead time may appear short it is significant both in minimizing the risk to life but also in allowing the first steps*

*to be taken in minimizing the risk to livelihoods.*

Page 14: Figure 3 is not easy to understand. What do you mean when you say "values of the time steps whose observed value exceeds the threshold"?
*The aim of this plot is to explore how the forecast performance changes with the magnitude of the forecast variable; in this case water level. The performance is assessed using the Continuous Rank Probability Score (CRPS) which is computed for each time step at each available lead time. Each line on the plot represents a different model/forecast lead time combination. For one such combination the values plotted to give the line are given by averaging the CRPS over different sets of forecasts. For example the value plotted for a threshold value of 8 corresponds to the average CPRS values for forecasts of observed values greater than 8m. The shorter vertical lines seen on the plot represent +/- 2 standard deviations of the estimate of the average value. The interpretation of this plot will be expanded upon with the paper.*

*Plots of this nature can be a useful diagnostic tool for ensuring that the identified model performs well at time of high water level when the forecasts are most important. For example in Fig. 3 the performance of model with the power law non-linearity is markedly better at high water levels (e.g. at a threshold value of 8m). This is not as apparent if the performance is computed over all the data (the lowest threshold shown 1m)*

Page 15: You demonstrate that the model would have accurately foreseen the crossing of the warning/danger levels five hours before the floods of 2013 and 2014. However, are there any other instances in the model hindcasts that would have unnecessarily crossed the danger level and given a false alarm? What is the probability of a false alarm?
*For the two illustrated events, the forecast registered a greater than 80% probability of a warning level being reached. With such a high probability an advance warning would likely be triggered. Comparing the forecast of these two events with the actual scenario, both events resulted in water levels reaching warning levels and a warning being triggered. Therefore neither of these forecasts would be considered a false alarm. There*

*were no other cases in the available data sample where a forecast gave a greater than 50% probability of exceeding the warning threshold, and therefore no potential false alarms were found in this sample. However the limited time period of data (2011-2015) and number of observed events (the two shown) means care should be taken in interpreting the level of discrimination between flood and non-flood events that may be available from the forecasts*

Page 15: How frequently do your forecast cross the danger level? How does this compare to the frequency of the danger level happening in real life?
*In the period 2011-2015 the observed data crossed the danger level twice, at the two time steps shown in Figures 4 & 5. These are the only periods where the model shows a high probability of exceeding the danger level.*

Page 15: In general, the accuracy of the model for low flows is not of particular interest in this case, as the goal is to provide early warnings for extreme floods. It would be of interest to the reader to have more statistics on the extreme events. What data is available for you to work with? Is it possible to create hindcasts of your model? If so, can you calculate the extent to which these forecasts would match up with the historical records from Desinventar?
*Available hourly data cover 2011 to 2015 (inclusive). Further periods could be hindcast if data were available but observed data prior to this time are recorded at a daily time step and therefore are not suitable for the model.*

Table 3 and Table 4 provide summary of the model performance during the calibration period, but it would also be good for the reader to see how each model performed during the test period (non-calibration period). How were the calibration periods and test periods selected?
*Appropriate columns can be added to the tables. The methodology for the selection of the periods is outlined on Pg 13 Lines 5-10. They are designed to mimic real life usage where the model has been revised at the start of each calendar year to incorporate the past year of data before forecasting the next monsoon period.*

Page 16 Line 5: Which non-linearity is being used, and why? How are they actually testing this? It would be of interest to include more details on this.

*They are using a Power law non-linearity. For this catchment the identification stage of the model development (reported in Section 5, Tables 3 &4 and Fig 3) suggests this is the best choice. More details of the testing will be included in the additional section of the paper to be added as mentioned earlier in this response and in response to RC1.*